# Structural insights into hormone recognition by the human glucose-dependent insulinotropic polypeptide receptor

Fenghui Zhao[1,2†], Chao Zhang[3,4†], Qingtong Zhou[5†], Kaini Hang[3†], Xinyu Zou[6], Yan Chen[1,2,5], Fan Wu[3], Qidi Rao[3,4], Antao Dai[7], Wanchao Yin[2], Dan-Dan Shen[8], Yan Zhang[8], Tian Xia[6], Raymond C Stevens[3], H Eric Xu[2,4*], Dehua Yang[2,4,7*], Lihua Zhao[2,4*], Ming-Wei Wang[1,2,3,4,5,7*]

[1]School of Pharmacy, Fudan University, Shanghai, China; [2]The CAS Key Laboratory of Receptor Research, Shanghai Institute of Materia Medica, Chinese Academy of Sciences, Shanghai, China; [3]School of Life Science and Technology, ShanghaiTech University, Shanghai, China; [4]University of Chinese Academy of Sciences, Beijing, China; [5]Department of Pharmacology, School of Basic Medical Sciences, Fudan University, Shanghai, China; [6]School of Artificial Intelligence and Automation, Huazhong University of Science and Technology, Wuhan, China; [7]The National Center for Drug Screening, Shanghai Institute of Materia Medica, Chinese Academy of Sciences, Shanghai, China; [8]Department of Biophysics and Department of Pathology of Sir Run Run Shaw Hospital, Zhejiang University School of Medicine, Hangzhou, China

*For correspondence:
eric.xu@simm.ac.cn (HEX);
dhyang@simm.ac.cn (DY);
zhaolihuawendy@simm.ac.cn (LZ);
mwwang@simm.ac.cn (M-WW)

†These authors contributed equally to this work

Competing interests: The authors declare that no competing interests exist.

**Abstract** Glucose-dependent insulinotropic polypeptide (GIP) is a peptide hormone that exerts crucial metabolic functions by binding and activating its cognate receptor, GIPR. As an important therapeutic target, GIPR has been subjected to intensive structural studies without success. Here, we report the cryo-EM structure of the human GIPR in complex with GIP and a $G_s$ heterotrimer at a global resolution of 2.9 Å. GIP adopts a single straight helix with its N terminus dipped into the receptor transmembrane domain (TMD), while the C terminus is closely associated with the extracellular domain and extracellular loop 1. GIPR employs conserved residues in the lower half of the TMD pocket to recognize the common segments shared by GIP homologous peptides, while uses non-conserved residues in the upper half of the TMD pocket to interact with residues specific for GIP. These results provide a structural framework of hormone recognition and GIPR activation.

## Introduction

Glucose-dependent insulinotropic polypeptide (GIP) is a 42-amino acid peptide hormone that plays crucial role in glucose regulation and fatty acid metabolism. In response to food intake, GIP is secreted by intestinal K cells to enhance insulin secretion and peripheral fatty acid uptake (*Kim et al., 2007*), as well as a number of neuronal effects (*Faivre and Hölscher, 2013*). The pleiotropic functions of GIP is mediated by its cognate receptor (GIPR), a member of class B1 G protein-coupled receptors (GPCRs) that also include glucagon receptor (GCGR) and glucagon-like peptide-1 receptor (GLP-1R). GIPR, together with GCGR and GLP-1R, forms the central endocrine network in regulating insulin sensitivity and energy homeostasis, and they are validated drug targets (*Lagerström and Schiöth, 2008*; *Finan et al., 2016*; *Longuet et al., 2008*). Intensive efforts were

made in drug discovery targeting these receptors (*Yang et al., 2021*). A number of GLP-1R selective ligands have been developed successfully to treat type 2 diabetes and obesity. Encouragingly, peptide ligands that bind both GIPR and GLP-1R show better clinical efficacy than the GLP-1R agonist alone. As such, GIPR has emerged as a hot target pursued by pharmaceutical research community.

GIPR contains a large extracellular domain (ECD) and a 7-transmembrane domain (TMD). Both are involved in ligand recognition and receptor activation (*Parthier et al., 2009*; *Koth et al., 2012*; *Yang et al., 2015*). Cryo-electron microscopy (cryo-EM) structures of GCGR and GLP-1R, as well as several other class B1 GPCRs have been solved, providing a general mechanism of two-domain model for peptide recognition and receptor activation. However, GIP displays an exquisite sequence specificity towards GIPR as it does not bind to other class B1 GPCRs. However, the efforts to understand the ligand selectivity by GIPR have been hampered by technical difficulties in expression and stabilization of the ligand-GIPR complexes for structural studies. We have overcome such challenges and determined a high-resolution (2.9 Å) structure of the human GIPR in complex with the stimulatory G protein ($G_s$) using single-particle cryo-EM approach in conjunction with NanoBiT strategy (*Duan et al., 2020*). Together with functional studies, our results demonstrate several unique structural features that distinguish GIPR from other members of the glucagon subfamily of class B1 GPCRs and provide an important template for rational design of GIPR agonists for therapeutic development.

## Results

### Structure determination

To prepare a high-quality human GIPR–$G_s$ complex, we overcame several technical obstacles to enhance the expression level and protein stability by adding a double tag of maltose binding protein at the C terminus and a BRIL fusion protein at the N terminus (*Figure 1—figure supplement 1A*), as well as employing the NanoBiT tethering strategy (*Duan et al., 2020*; *Zhou et al., 2020*; *Sun et al., 2020*; *Figure 1—figure supplement 1A,B*). To solve the $GIP_{1-42}$–GIPR–$G_s$ structure, we further introduced one mutation (T345F) to stabilize the assembly of complex (*Figure 1—figure supplement 1C,D*). This mutation does not affect the ligand binding or potency of $GIP_{1-42}$ in cAMP accumulation assay (*Figure 1—figure supplement 1G,H*). Large-scale purification was followed and the $GIP_{1-42}$–GIPR–$G_s$ complexes were collected by size-exclusion chromatography (SEC) for cryo-EM studies (*Figure 1—figure supplement 1E,F*). The activity of the modified GIPR construct was confirmed by cAMP accumulation assay showing a response similar to that of the wild-type (WT; *Figure 1—figure supplement 1G*).

The $GIP_{1-42}$–GIPR–$G_s$ complexes were imaged using a Titan Krios equipped with a Gatan K3 Summit direct electron detector (*Figure 1—figure supplement 2*). 2D classification showed a clear secondary structure feature and random distribution of the particles. Different directions of the particles enabled a high-resolution cryo-EM map reconstruction (*Figure 1—figure supplement 2B*). A total of 295,021 particles were selected after 3D refinement and polishing, leading to an overall resolution of 2.9 Å (*Figure 1—figure supplement 2C,D* and *Table 1*).

### Overall structure

Apart from the α-helical domain (AHD) of $Gα_s$ which is flexible in most cryo-EM GPCR–G protein complex structures, the bound $GIP_{1-42}$, GIPR, and $G_s$ were well defined in the EM density maps (*Figure 1*, *Figure 1—figure supplement 3*). Except for the ECD, side chains of the majority of amino acid residues are well resolved in all protein components. The final model contains 30 $GIP_{1-42}$ residues, the Gαβγ subunits of $G_s$, and the GIPR residues from $Q30^{ECD}$ to $S415^{8.66b}$ (class B GPCR numbering in superscript) (*Wootten et al., 2013*), with six amino acid residues missing at helix 8. As a general feature in most reported class B1 GPCR–$G_s$ complex structures (*Qiao et al., 2020*; *Zhang et al., 2017a*; *Zhao et al., 2019*; *Ma et al., 2020*), the density of ECD is relatively poor owing to its intrinsic flexibility, which limited the accuracy in model building for the GIPR ECD region compared to other regions of the complex structure. Given a low resolution of the density map, the ECD structure is model based on the crystal structure of GIPR ECD (PDB code: 2QKH). Notable conformation difference from GCGR (*Qiao et al., 2020*) or GLP-1R (*Zhang et al., 2017a*) was observed in the extracellular loop 1 (ECL1).

**Table 1.** Cryo-EM data collection, refinement, and validation statistics.

**GIP–GIPR–G$_s$–Nb35 complex**

| Data collection and processing | |
|---|---|
| Magnification | 46,685 |
| Voltage (kV) | 300 |
| Electron exposure (e$^-$/Å$^2$) | 80 |
| Defocus range (μm) | −1.2 to −2.2 |
| Pixel size (Å) | 1.071 |
| Symmetry imposed | C1 |
| Initial particle images (no.) | 4,895,399 |
| Final particle images (no.) | 295,021 |
| Map resolution (Å)<br>FSC threshold | 2.9<br>0.143 |
| Map resolution range (Å) | 2.7–5.0 |
| | |
| Refinement | |
| Initial model used (PDB code) | PDB codes 6WPW and 2QKH |
| Model resolution (Å)<br>FSC threshold | 2.9<br>0.5 |
| Model resolution range (Å) | 2.7–5.0 |
| Map sharpening B factor (Å$^2$) | −86.3 |
| Model composition<br>Non-hydrogen atoms<br>Protein residues<br>Lipids | <br>9409<br>1156<br>6 |
| B factors (Å$^2$)<br>Protein<br>Ligand<br>Lipids | <br>133<br>143<br>121 |
| R.m.s. deviations<br>Bond lengths (Å)<br>Bond angles (Å) | <br>0.005<br>1.036 |
| Validation<br>MolProbity score<br>Clash score<br>Poor rotamers (%) | <br>1.21<br>4.23<br>0.00 |
| Ramachandran plot<br>Favored (%)<br>Allowed (%)<br>Disallowed (%) | <br>98.15<br>1.85<br>0.00 |

Similar to other class B1 GPCR–G$_s$ complexes, the TM6 of GIPR shows a sharp kink in the middle and TM7 displays an outward movement. Like parathyroid hormone receptor-1 (PTH1R)–G$_s$ and corticotropin-releasing factor receptor type 1 (CRF1R)–G$_s$ cryo-EM structures (*Zhao et al., 2019*; *Ma et al., 2020*), the TMD of GIPR is surrounded by annular detergent micelle, with a diameter of 12 nm thereby mimicking the lipid bilayer morphology (*Figure 1*). In addition, we also observed several cholesterols molecules in the cryo-EM map.

## Ligand recognition

In the complex, GIP adopts a single continuous helix that penetrates into the TMD core through its N-terminal half (residues 1–15), while the C-terminal half (residues 16–30) is recognized by the ECD and ECL1 (*Figure 2A–C*). Y1$^P$ (P indicates that the residue belongs to the peptide ligand) of GIP points to TMs 2–3, forms hydrogen bonds with R190$^{2.67b}$ and Q224$^{3.37b}$, and makes hydrophobic contacts with V227$^{3.40b}$ and W296$^{5.36b}$. This observation received support of the mutagenesis study,

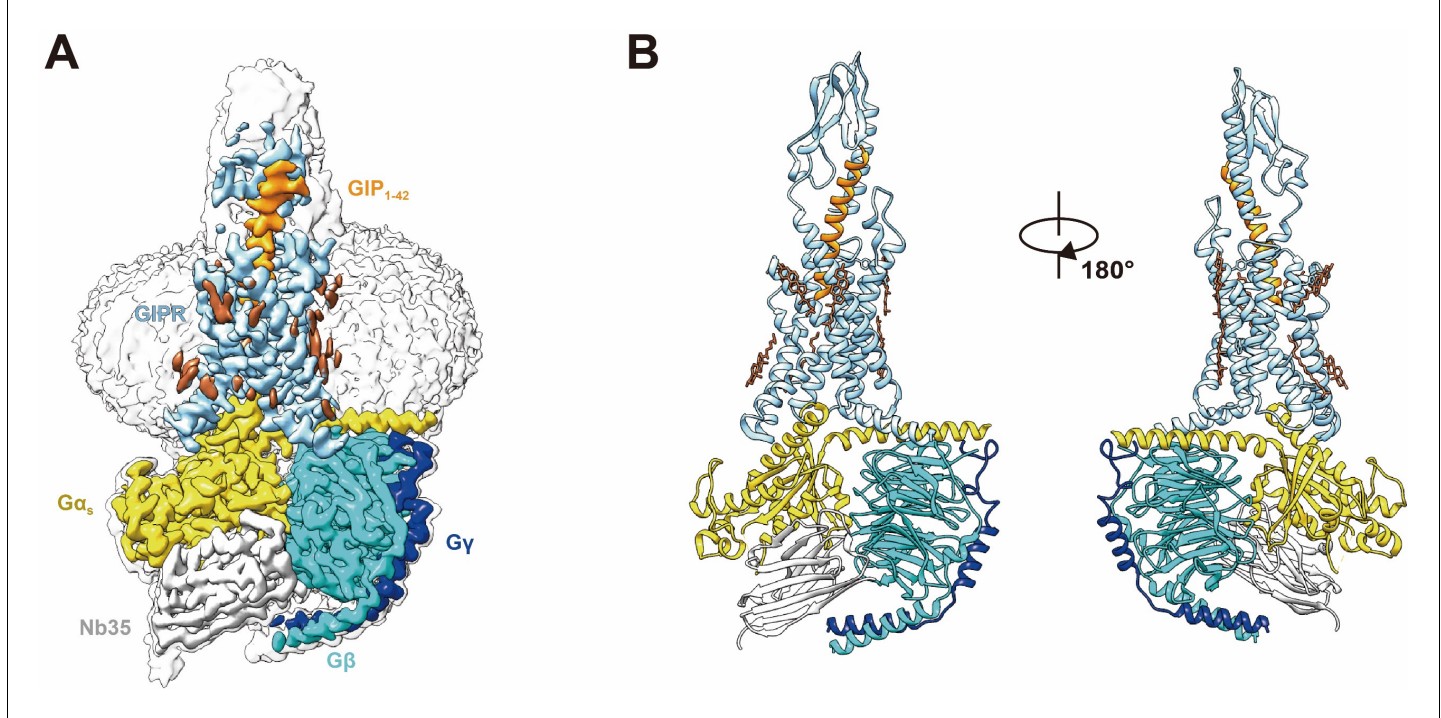

**Figure 1.** Cryo-EM structure of the GIP$_{1-42}$–GIPR–G$_s$ complex. (**A**) Cut-through view of the cryo-EM density map that illustrates the GIP$_{1-42}$–GIPR–G$_s$ complex and the disc-shaped micelle. The unsharpened cryo-EM density map at the 0.07 threshold shown as light gray surface indicates a micelle diameter of 11 nm. The colored cryo-EM density map is shown at the 0.16 threshold. (**B**) Model of the complex as a cartoon, with GIP$_{1-42}$ as helix in orange. The receptor is shown in light sky blue, G$\alpha_s$ in yellow, G$\beta$ subunit in cyan, G$\gamma$ subunit in navy blue, and Nb35 in gray.

The online version of this article includes the following source data and figure supplement(s) for figure 1:

**Source data 1.** Effects of GIP$_{1-42}$-mediated cAMP accumulation and binding affinity.

**Figure supplement 1.** Purification and characterization of the GIP$_{1-42}$–GIPR–G$_s$–Nb35 complex.

**Figure supplement 2.** Cryo-EM analysis of the GIP$_{1-42}$–GIPR–G$_s$ complex.

**Figure supplement 3.** Atomic resolution model of the GIP$_{1-42}$–GIPR–G$_s$ complex in the cryo-EM density map.

where mutant W296A decreased the potency of GIP-induced cAMP signaling by 50-fold (*Figure 2D*), and the reductions in mutants R190A and Q224A were 71- and 5-fold, respectively, as reported in a previous report (*Yaqub et al., 2010*). N-terminal truncation of either Y1$^P$ or both Y1$^P$ and A2$^P$ led to reduced efficacy or loss of activity (*Kerr et al., 2011*; *Gabe et al., 2020*), highlighting a crucial role of Y1$^P$. E3$^P$, D9$^P$, and D15$^P$ are three negatively charged residues in the N-terminal half of GIP and form salt bridges with R183$^{2.60b}$, R370$^{7.35b}$, and R289$^{ECL2}$, respectively. Removal of these salt bridges by alanine substitution at either R183$^{2.60b}$ (*Yaqub et al., 2010*) or R370$^{7.35b}$ (*Figure 2D*) greatly reduced GIP potency (by 76- and 55-fold, respectively), whereas the effect on mutant R289A was mild (6-fold, *Figure 2D*). Polar interactions also occurred between S8$^P$ and N290$^{ECL2}$ as well as Y10$^P$ and Q138$^{1.40b}$. The GIP–TMD interface was further stabilized by a complementary nonpolar network involving TM1 (L134$^{1.36b}$, L137$^{1.39b}$, and Y141$^{1.43b}$) and TM7 (L374$^{7.39b}$ and I378$^{7.43b}$) via A2$^P$, F6$^P$, and Y10$^P$ of GIP (*Figure 2C*), in line with decreased ligand potencies observed in Y141A (by 103-fold), L374A (by 41-fold), and I378A (by 8-fold) mutants (*Figure 2D*). These mutants also caused significant potency decreases in GIP$_{1-42}$-induced β-arrestin2 recruitment (*Figure 2—figure supplement 1*).

The C-terminal half of GIP was clasped by the GIPR ECD, closely resembling the crystal structure of GIP–GIPR ECD (PDB code: 2QKH) (*Parthier et al., 2007*). Consistent with the interaction patterns observed in other class B1 GPCRs (*Parthier et al., 2007*), the hydrophobic residues (F22$^P$, V23$^P$, L26$^P$, and L27$^P$) in the C-terminal half of GIP occupy a complementary binding groove of the GIPR ECD, consisting of a series of hydrophobic residues (L35, Y36, W39, M67, Y68, Y87, L88, P89, and W90). Alanine substitutions in W39, D66, and Y68 significantly reduced the potency of GIP

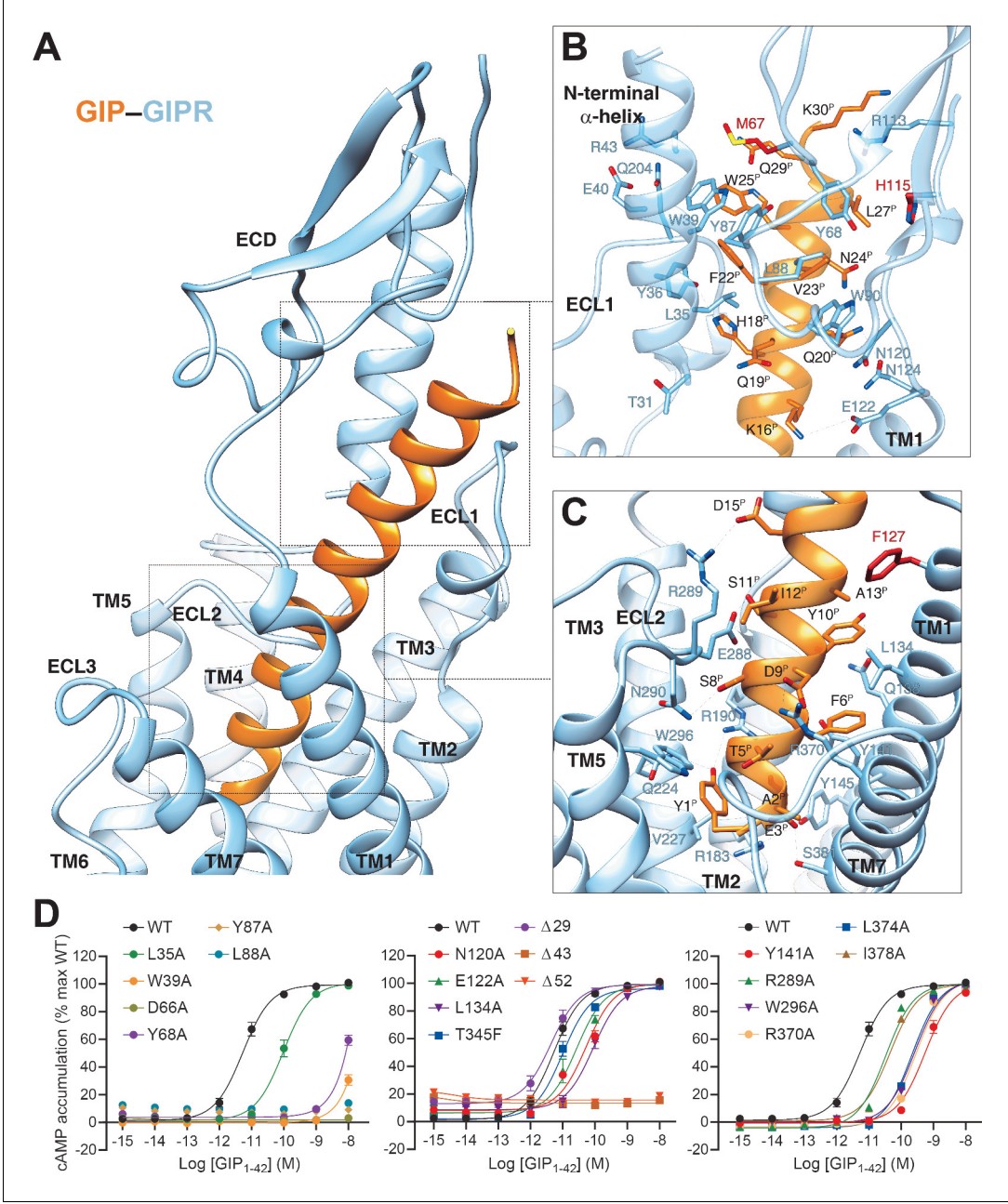

**Figure 2.** Molecular recognition of GIP by GIPR. (**A**) The binding mode of GIP (orange) with GIPR (light sky blue), showing that the N-terminal half of GIP penetrates into a pocket formed by all TM helices except TM4, ECL2, and ECL3, whereas the C-terminal half is recognized by ECD, ECL1, and TM1. (**B, C**) Close-up views of the interactions between GIP and GIPR. The residues and side chains that could not be modelled in the ECD are colored in red. (**D**) Signaling profiles of GIPR mutants. cAMP accumulation in wild-type (WT) and single-point mutated GIPR expressing in HEK 293T cells. Signals were normalized to the maximum response of the WT and dose–response curves were analyzed using a three-parameter logistic equation. All data were generated and graphed as means ± S.E.M. of at least three independent experiments, conducted in quadruplicate. Δ, truncated residues.

The online version of this article includes the following source data and figure supplement(s) for figure 2:

**Source data 1.** Effects of residue mutation in the ligand-binding pocket on GIP$_{1-42}$-induced cAMP accumulation, cell surface expression, and binding affinity.

**Figure supplement 1.** Effects of residue mutation in the ligand-binding pocket on GIP$_{1-42}$-induced β-arrestin2 recruitment.

**Figure supplement 2.** Conformational changes upon GIPR activation.

(*Figure 2D*). Besides, several polar contacts including H18$^P$-Y36 and Q20$^P$-N124 were observed. Notably, the cryo-EM map suggests that the ECL1 stands upwards to approach the N-terminal α-helix of ECD and forms hydrogen bonds with the side chain of Y36 (*Figure 2A,B*), resulting in a close contact between TMD and ECD for GIP-bound GIPR (interface area = 571 Å$^2$), significantly larger than that of GLP-1-bound GLP-1R (362 Å$^2$), reinforcing the importance of ECD in GIP recognition.

## Receptor activation

GIPR shares ~50% sequence similarity with GCGR, especially in the TMD region (75%); thus, GCGR structures published previously provide a good template for the present study (*Figure 2—figure supplement 2*; *Qiao et al., 2020*; *Hilger et al., 2020*; *Jazayeri et al., 2016*; *Zhang et al., 2018*; *Chang et al., 2020*). It was found the TMD of activated GIPR exhibits a conformation similar to that of GCGR activated by glucagon or ZP3780 (Cα RMSD = 1.2 and 0.7 Å, respectively) (*Qiao et al., 2020*; *Hilger et al., 2020*) and distinct from that of GCGR bound by the negative allosteric modulator NNC0640 or partial agonist NNC1702 (Cα RMSD = 4.0 and 3.9 Å, respectively) (*Zhang et al., 2017b*). Facilitated by Gly$^{7.50b}$ located in the middle of TM7, the extracellular half of TM7 bends towards TM6 by 8.0 Å (measured by Cα atom of Gly$^{7.32b}$) (*Figure 2—figure supplement 2*). This feature and the outward movement of ECL3 expanded the ligand binding pocket. Meanwhile, the extracellular tip of TM1 was extended by one turn and moved inward by 8.0 Å (measured by Cα atom of the residues at 1.30b) (*Figure 2—figure supplement 2*). Together with the raised ECL1, these conformational changes stabilized ligand binding.

In the intracellular side, the sharp kink in the middle of TM6 led to an outward movement of its intracellular portion measured by Cα atom of R336$^{6.35b}$ (18.9 Å, similar to that of other G$_s$-coupled class B1 receptors). This was accompanied by the movement of the intracellular tip of TM5 toward TM6 by 7.6 Å (measured by Cα atom of the residues at 5.67b), thereby creating an intracellular cavity for G protein coupling (*Figure 2—figure supplement 2*).

## G protein coupling

In our model, G$_s$ protein is anchored by the α5 helix of Gα$_s$ (GαH5), thereby fitting to the cytoplasmic cavity formed by TMs 3, 5, and 6, intracellular loops (ICLs) 1–2 and H8 (*Figure 3*). In general, the GIPR–G$_s$ complex shows a similar receptor–G protein interface as other reported class B1 receptor structures such as GLP-1R (*Zhang et al., 2020*), GLP-2R (glucagon-like peptide-2 receptor) (*Sun et al., 2020*), GCGR (*Qiao et al., 2020*), PTH1R (*Zhao et al., 2019*), SCTR (secretin receptor) (*Dong et al., 2020*), and GHRHR (growth hormone-releasing hormone receptor) (*Zhou et al., 2020*), suggesting a common G protein signaling mechanism (*Figure 3A*). The hydrophobic residues at the C-terminal of GαH5 (L388$^{GαH5}$, Y391$^{GαH5}$, L393$^{GαH5}$, and L394$^{GαH5}$) insert into a small hydrophobic pocket formed by Y240$^{3.53b}$, L241$^{3.54b}$, L244$^{3.57b}$, L245$^{3.58b}$, I317$^{5.58b}$, I320$^{5.60b}$, L321$^{5.61b}$, and L325$^{5.65b}$ (*Figure 3B*). The side chain of R338$^{6.37b}$ points to Gα$_s$ and makes one hydrogen bond with L394$^{GαH5}$. Of note is that the interaction between R380$^{GαH5}$ and ICL2 results in five hydrogen bonds with the backbone atoms of L245$^{3.58b}$, V246$^{3.59b}$, L247$^{3.60b}$, and V248$^{ICL2}$, significantly more than that observed in GLP-1R, SCTR, or GCGR (*Figure 3C*). The polar residues in ICL2 (S251$^{ICL2}$ and E253$^{ICL2}$) produce two hydrogen bonds with K34 and Q35 of Gα$_s$, while H8 forms several hydrogen bonds with ICL1, then contacts with Gβ (E398$^{8.49b}$-R164$^{ICL1}$-D312$^{Gβ}$, E402$^{8.53b}$-R164$^{ICL1}$-D312$^{Gβ}$) (*Figure 3D*). Together, these specific interactions contribute to the G$_s$ coupling specificity of GIPR.

## Ligand specificity

GIP, GLP-1, and glucagon are three important metabolic hormones exerting distinct functions in glucose homeostasis, in spite of high degrees of sequence similarity. Superimposing the TMD of GIP-bound GIPR with that of GLP-1-bound GLP-1R (*Zhang et al., 2020*) or glucagon-bound GCGR (*Qiao et al., 2020*) displays a similar ligand-binding pocket and the three peptides all adopt a single continuous helix, with the N terminus penetrating to the TMD core to the same depth, while the C terminus anchors the ECD and ECL1 in a receptor-specific manner (*Figure 4*). Notably, the ECL1 of GIPR stands upwards in line with TMs 2 and 3 and moves towards the TMD core by 5~7 Å. Such a movement, together with a α-helical extension in TM1 by six residues, allows GIP to shift to TM1 by 2.7 and 3.3 Å (measured by Cα atom of L27$^P$) relative to GLP-1 (*Zhang et al., 2020*) and glucagon (*Qiao et al., 2020*), respectively (*Figure 4—figure supplement 1*).

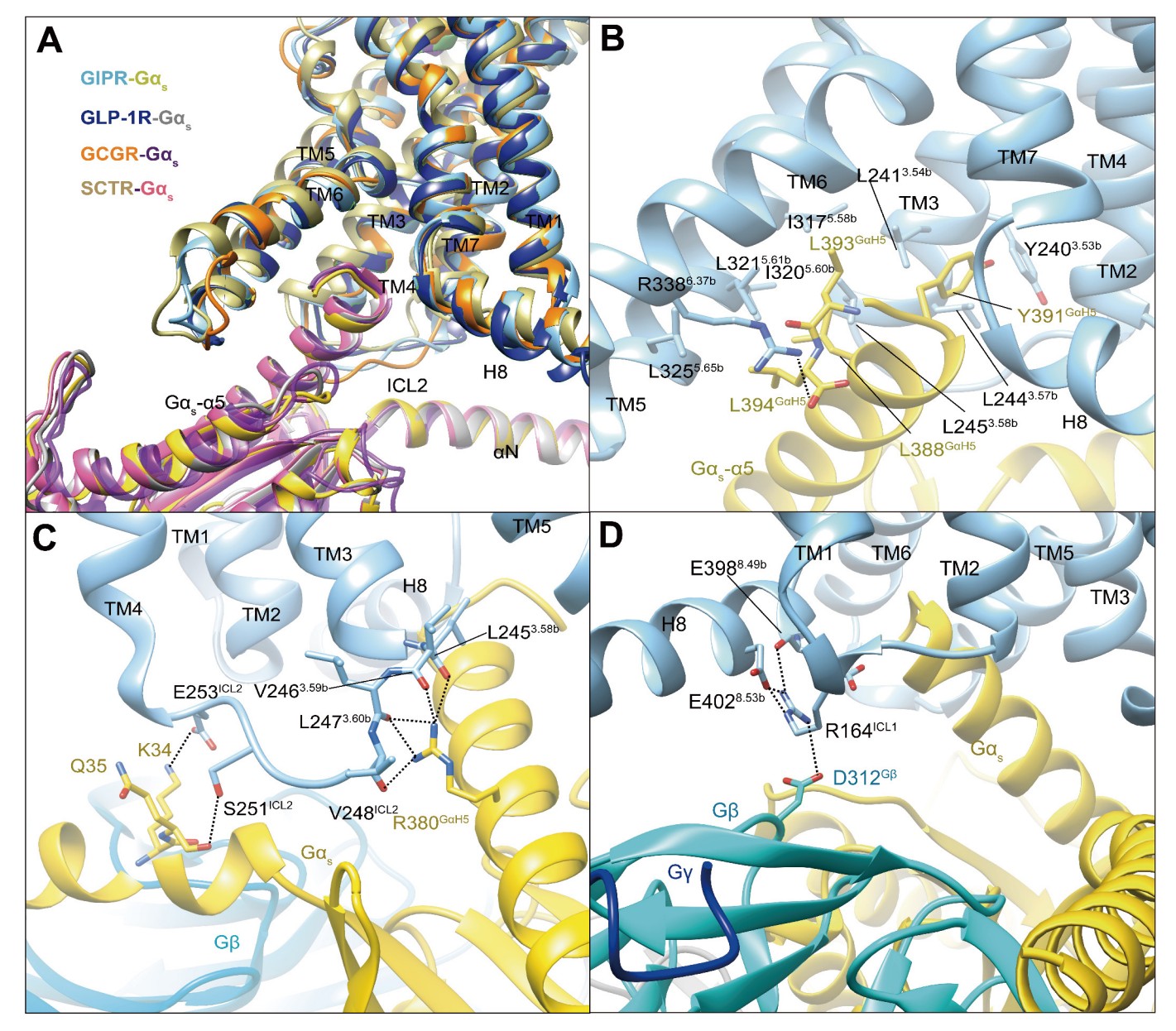

**Figure 3.** G protein coupling of GIPR. (**A**) Comparison of G protein coupling among GIPR, GLP-1R (*Zhang et al., 2020*), GCGR (*Qiao et al., 2020*), and SCTR (*Dong et al., 2020*). The Gα$_s$ α5-helix of the Gα$_s$ Ras-like domain inserts into an intracellular crevice of GIPR TMD. (**B**) Interaction between GIPR and the C terminus of Gα$_s$. (**C**) Polar interactions between ICL2 and Gα$_s$. (**D**) Polar interactions between H8 and ICL1 of the GIPR and Gβ. The GIP$_{1-42}$–GIPR–Gα$_s$ structure is colored light sky blue (GIPR), gold (Gα$_s$), and cyan (Gβ). Residues involved in interactions are shown as sticks. Polar interactions are shown as black dashed lines.

Based on the sequence similarity, the three peptides can be divided into four segments: two common segments (residues 4–11 and 21–30 in GIP) and two unique segments (residues 1–3 and 12–20 in GIP) (*Figure 4H*). The N terminus (residues 1–3) makes massive contacts with the conserved central polar network of class B1 GPCRs including one hydrogen bond with Q$^{3.37b}$ stabilized by the hydrophobic residue at 3.40b; one hydrogen bond with Y$^{1.47b}$ made by the third peptide residue (*Figure 4B,H*); residues 4–11 interact with salt bridges of R$^{7.35b}$, pi-stacking of Y$^{1.43b}$, hydrophobic L$^{2.71b}$, W$^{5.36b}$, and L$^{7.39b}$, as well as several hydrogen bonds in ECL2 (*Figure 4C,H*); residues 12–20 are divergent and mainly interact with ECLs 1–2 and TMs 1–2 (*Figure 4D,H*).

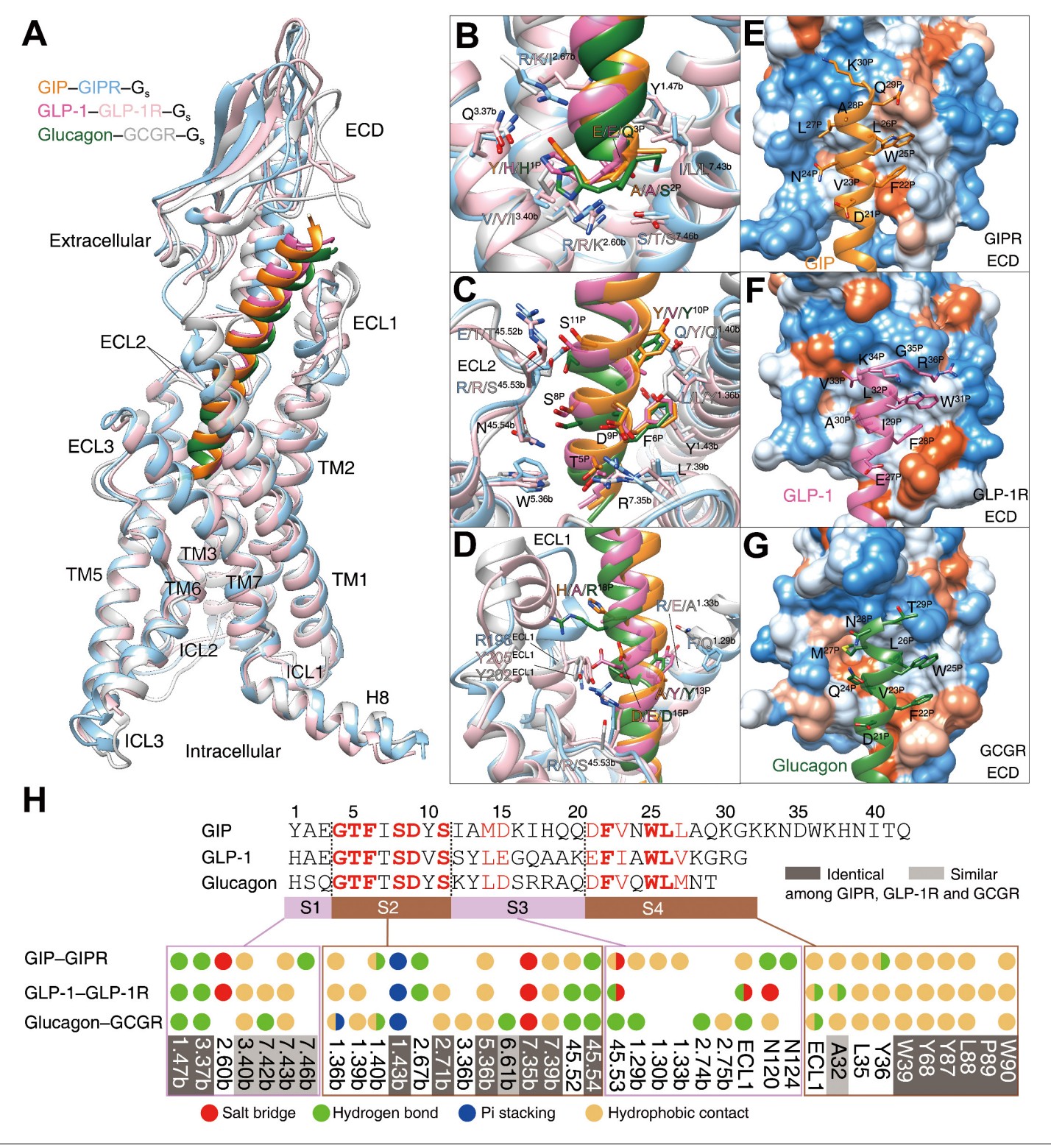

**Figure 4.** Ligand specificity among GIPR, GLP-1R, and GCGR. (**A**) Comparison of the overall structures of GIP₁₋₄₂–GIPR–G$_s$, GLP-1–GLP-1R–G$_s$ (*Zhang et al., 2020*) and glucagon–GCGR–G$_s$ complexes (*Qiao et al., 2020*). G proteins are omitted for clarity. (**B–D**) Close-up views of the interaction between TMD and peptide. Based on sequence similarity, the peptides are divided into four segments: N terminus (residues 1–3, **B**), segment 2 (residues 4–11, **C**), segment 3 (residues 12–20, **D**), and the C terminus (residues 21 to the end, **E–G**), where segments 2 and 4 are highly conserved among GIP, GLP-1, and glucagon. Residues are numbered based on GIP for peptides, and labeled with class B GPCR numbering in superscript for

*Figure 4 continued on next page*

*Figure 4 continued*

receptors (*Wootten et al., 2013*). (**E–G**) Close-up views of the interface between GIPR ECD and GIP C terminus (**E**), between GLP-1R and GLP-1 C terminus (**F**), and between GCGR and glucagon C terminus (**G**). The ECD is shown in surface representation and colored from dodger blue for the most hydrophilic region, to white, to orange red for the most hydrophobic region. (**H**) Comparison of peptide recognition modes for three receptors, described by fingerprint strings encoding different interaction types of the surrounding residues in each receptor. Peptide residue numbers on the top are shown based on GIP. The ligand-binding pocket residues that are identical or similar across three receptors are highlighted in dark gray and light gray, respectively. Color codes are listed on the bottom.

The online version of this article includes the following figure supplement(s) for figure 4:

**Figure supplement 1.** Structural comparison of ECL1 conformations among GIPR, GLP-1R and GCGR.
**Figure supplement 2.** Molecular dynamics (MD) simulations of GIPR in complex with GLP-1 and glucagon.
**Figure supplement 3.** Comparison of the cryo-EM structure of GIPR with other class B1 receptors.
**Figure supplement 4.** GIP$_{1-42}$, GLP-1$_{7-36}$, or glucagon-elicited cAMP accumulation was measured in HEK 293T cells expressing GIPR, GLP-1R, or GCGR, respectively.

To accommodate varying lengths of side chains at A$^{13P}$/Y/Y, I$^{17P}$/Q/R, Q$^{19P}$/A/A, and Q$^{20P}$/K/Q, both TM1 and ECL1 adjusted their conformations to avoid clashes (*Figure 4D, Figure 4—figure supplement 1*). For example, ECL1 of GLP-1R is more distant from GLP-1 than that of GIPR from GIP, whereas repulsion of the side chain of R$^{18P}$ was seen between GCGR and glucagon. Therefore, receptor-specific interaction may reside in this region, which precludes the binding of GLP-1 or glucagon to GIPR revealed by MD simulations (*Figure 4—figure supplement 2*). As far as C terminus is concerned, all three peptides form extensive hydrophobic contacts with the ECD, resulting from the hydrophobic composition of amino acids in both sides (*Figure 4E–H*). It appears that GIPR, GLP-1R, and GCGR employ conserved residues to recognize the common segments of their endogenous peptides and use non-conserved residues to make specific interaction that govern the ligand selectivity.

## Discussion

As one of the incretin hormones, GIP modulates glucose metabolism by stimulating the β-cells to release insulin (*Seino et al., 2010*). Unlike GLP-1, it does not suppress gastric emptying and appetite, while exerting opposite actions on pancreatic α-cells as well as adipocytes leading to glucagon secretion and lipogenesis (*Seino et al., 2010*). Coupled with reduced sensitivity in type 2 diabetic patients, development of GIPR-based therapeutics met little success (*Coskun et al., 2018*).

Comparison of the full-length structures of six glucagon subfamily of GPCRs demonstrates that bound peptides (GLP-1, exendin-P5, glucagon, ZP3780, secretin, GHRH, GLP-2, and GIP) all adopt a single straight helix with their N terminus inserted into the TMD core, while the C-terminal is recognized by the ECD (*Zhou et al., 2020*; *Sun et al., 2020*; *Qiao et al., 2020*; *Zhang et al., 2017a*; *Dong et al., 2020*). For parathyroid hormone subfamily of GPCRs, the long-acting PTH analog (LA-PTH) predominantly exhibits an extended helix with its N terminus inserted deeply into the TMD, where the peptide C terminus may bend occasionally (*Zhao et al., 2019*). In the case of CRF subfamily of GPCRs, the N terminus (first seven residues) of urocortin 1 (UCN1) and CRF1 present an extended loop conformation, and its C-terminal residues (8–40) adopt a single extended helix (*Ma et al., 2020*; *Liang et al., 2020*). As far as calcitonin subfamily of GPCRs is concerned, calcitonin gene-related peptide (CGRP) has an unstructured loop in both N- and C-terminal regions (*Liang et al., 2018a*). Looking at pituitary adenylate-cyclase-activating peptide (PACAP) and vasoactive intestinal polypeptide (VIP) receptor subfamily, PACAP displays an extended α-helix, while maxadilan, a natural PAC1R agonist (61-amino acid long), forms the N- and C-terminal helices that are linked as a loop (*Duan et al., 2020*; *Liang et al., 2020*; *Wang et al., 2020*). These observations highlight diversified peptide binding modes among class B1 GPCRs (*Figure 4—figure supplement 3*).

Species differences in class B1 receptor responsiveness are diversified and receptor specific, which is tolerable for some receptors such as GLP-1R and GCGR, but leads to concerns for others like GIPR and parathyroid hormone receptor-2 (PTH2R) (*Sparre-Ulrich et al., 2016*; *Hoare et al., 1999*). Interestingly, the sequence identities between human and mouse at both ligand and receptor levels are more conserved between GLP-1 and GLP-1R (100% and 93%) than that between GIP and GIPR (92% and 81%) (*Sparre-Ulrich et al., 2016*). Such a divergence is not caused by changes in

peptide potency, but resides in the biological property of either GIP or the receptor (*Sparre-Ulrich et al., 2016*; *Bailey, 2020*). However, it may affect GIP-related pharmacology markedly (*Sparre-Ulrich et al., 2016*). Indeed, a previous study found that human GIP is a comparatively weak partial agonist in rodent models (*Sparre-Ulrich et al., 2016*). Human (Pro3)GIP is a full agonist with identical maximum response as human GIP, whereas both rat and mouse (Pro3)GIPs are partial agonists (*Sparre-Ulrich et al., 2016*; *Bailey, 2020*). Of note is that among rat, mouse, and human GIPs, the only residue change (from His to Arg) occurs at the 18th position (*Sparre-Ulrich et al., 2016*). From a structural biology perspective, the variation in the sequences of both GIP (H[18P]/R/R for human, rat, and mouse) and GIPR, and the consequent alterations in either peptide-binding or G protein-coupling may offer an explanation. Nonetheless, it may also complicate knowledge transfer from rodents to humans for clinical development of GIPR-based therapeutics.

The interactions between the three receptors (GIPR, GLP-1R, and GCGR) and their endogenous peptides transduce precise cellular signals responsible for glucose control. While $GIP_{1-42}$, GLP-1, and glucagon each binds to the cognate receptor with high affinity ($pIC_{50}$ = 8.07, 8.25 and 7.31, respectively), glucagon also cross-reacts with GLP-1R with a $pIC_{50}$ value of 6.19 (*Yuliantie et al., 2020*; *Darbalaei et al., 2020*). This property is consistent with their behavior in inducing cAMP responses: $GIP_{1-42}$ and GLP-1 specifically activate GIPR and GLP-1R, respectively, whereas glucagon can elicit cAMP accumulation mediated by both GCGR and GLP-1R ($EC_{50}$ = 1.14 nM; *Figure 4—figure supplement 4*), highlighting the complexity of their interactive functionalities. Our studies show that the recognition pattern among these three peptide–receptor pairs is instituted by a common and closely related mechanism where the extracellular portion of the receptor mainly binds to a cognate ligand, while the TMD activates a cascade of signaling events. The upper half of the TMD pocket composed of the top parts of ECL1, TM1, and TM2 interacts with unique residues in the peptide through flexible movement of ECL1 and complementary shape formation by TM1 and TM2, thereby conferring selectively and discriminating unrelated ligands. The lower half of the TMD pocket composed of TMs 3, 6, and 7 displays conserved sequences for recognition of common residues in the peptide. Its key function is to converge external signal into the cytoplasm and executes transduction with high efficiency. This mechanistic design reflects evolutionary advantages because multiple polypeptides could be accurately recognized via different sequences in the upper half of the TMD pocket.

Finally, GIPR, combined with GLP-1R and GCGR, have been intensively studied as targets of dual- or tri- agonists (*Skow et al., 2016*; *Alexiadou et al., 2019*). Combined activation of GLP-1R and GIPR by dual agonists would provide synergistic and improved effects in glycemic and body weight control (*Bastin and Andreelli, 2019*). The GLP-1R/GIPR dual-agonists LY3298176 (developed by Eli Lilly) and NN9709 (developed by Novo Nordisk/Marcadia) as well as GLP-1R/GCGR/GIPR tri-agonist HM15211 (developed by Hamni Pharmaceuticals) are undergoing phase II or III clinical trials (*Yang et al., 2021*). The detailed structural information on GIPR reported here will certainly be of value to better understand the mode of actions of these therapeutic peptides.

## Materials and methods

### Key resources table

| Reagent type (species) or resource | Designation | Source or reference | Identifiers | Additional information |
|---|---|---|---|---|
| Gene | GIPR_human | NCBI | NM_000164.4 | |
| Strain, strain background (*Escherichia coli*) | BL21 (DE3) | TIANGEN | Cat # CB105 | |
| Cell line (*Homo sapiens*) | HEK 293T | ATCC | Cat # CRL-3216 | |
| Cell line (hamster) | CHO-K1 | ATCC | Cat # CCL-61 | |
| Cell line (insect) | *Sf9* | Invitrogen | N/A | |
| Cell line (insect) | High-Five insect cells | ThermoFisher Scientific | Cat # B85502 | |

*Continued on next page*

*Continued*

| Reagent type (species) or resource | Designation | Source or reference | Identifiers | Additional information |
|---|---|---|---|---|
| Recombinant DNA reagent | pFastBac-HA-BRIL-TEV -2GSA-GIPR(22-421) T345F-15AA-LgBiT-TEV-OMBP-MBP | This paper | N/A | |
| Recombinant DNA reagent | pFastBac-HA-BRIL-TEV-2GSA-GIPR (22-421)−15AA-LgBiT-TEV-OMBP-MBP | This paper | N/A | |
| Recombinant DNA reagent | pFastBac-DNGα$_s$ | This paper | N/A | |
| Recombinant DNA reagent | pFastBac-Gβ1-peptide 86 | https://doi.org/10.1038/s41422-020-00442-0 | N/A | |
| Recombinant DNA reagent | pFastBac-Gγ2 | https://doi.org/10.1038/s41422-020-00442-0 | N/A | |
| Recombinant DNA reagent | PMESy4-Nb35 | https://doi.org/10.1016/j.molcel.2020.01.013 | N/A | |
| Recombinant DNA reagent | pcDNA3.1-GIPR (WT and mutants)−3Flag | This paper | N/A | |
| Peptide, recombinant protein | GIP$_{1-42}$ | GenScript | N/A | |
| Chemical compound, drug | Protease Inhibitor Cocktail, EDTA-Free | TragetMol | Cat # C0001 | |
| Chemical compound, drug | Apyrase | Sigma-Aldrich (Merck) | Cat # A6132 | |
| Chemical compound, drug | TCEP | Sigma-Aldrich (Merck) | Cat # C4706 | |
| Chemical compound, drug | Lauryl maltose neopentylglycol (LMNG) | Anatrace | Cat # NG310 | |
| Chemical compound, drug | Cholesterol hemisuccinate (CHS) | Anatrace | Cat # CH210 | |
| Chemical compound, drug | Glyco-diosgenin (GDN) | Anatrace | Cat # GDN101 | |
| Chemical compound, drug | Amylose resin | NEB | Cat # E8021L | |
| Chemical compound, drug | ESF 921 culture medium | Expression Systems | Cat # 96-00-01 | |
| Chemical compound, drug | Fetal bovine serum (FBS) | Gibco | Cat # 10099–141 | |
| Chemical compound, drug | DMEM | Gibco | Cat # 12430–054 | |
| Chemical compound, drug | X-tremeGHNE HP DNA Transfection Reagent | Sigma-Aldrich (Roche) | Cat # 6366236001 | |
| Chemical compound, drug | Digitonin | Biosynth | Cat # D-3203 | |
| Chemical compound, drug | Salt active nuclease | Sigma-Aldrich | Cat # SRE0015-5KU | |
| Chemical compound, drug | Sodium pyruvate | Gibco | Cat # 11360–0'70 | |
| Chemical compound, drug | Lipofectamine 2000 transfection reagent | Invitrogen | Cat # 11668–019 | |
| Chemical compound, drug | $^{125}$I-GIP | PerkinElmer | Cat # NEX402010UC | |

*Continued on next page*

*Continued*

| Reagent type (species) or resource | Designation | Source or reference | Identifiers | Additional information |
|---|---|---|---|---|
| Chemical compound, drug | BSA | ABCONE | Cat # A23088-100G | |
| Antibody | Anti-Flag primary antibody | Sigma-Aldrich | Cat # F3165 | |
| Antibody | Anti-mouse Alexa Fluor 488 conjugated secondary antibody | Invitrogen | Cat # A-21202 | |
| Commercial assay, kit | LANCE Ultra cAMP kit | PerkinElmer | Cat # 2675984 | |
| Software, algorithm | MotionCor2.1 | doi:10.1126/science.aav7942 | N/A | https://msg.ucsf.edu/em/software/motioncor2.html |
| Software, algorithm | Gctf v1.06 | https://doi.org/10.1016/j.jsb.2015.11.003 | N/A | https://www2.mrc-lmb.cam.ac.uk/research/locally-developed-software/zhang-software/ |
| Software, algorithm | RELION-3.0-beta2 | https://doi.org/10.1016/j.jsb.2012.09.006 | N/A | https://www3.mrc-lmb.cam.ac.uk/relion/index.php/Download_%26_install |
| Software, algorithm | COOT | https://doi.org/10.1107/S0907444904019158 | N/A | https://www2.mrc-lmb.cam.ac.uk/personal/pemsley/coot/ |
| Software, algorithm | Phenix | https://doi.org/10.1107/S0907444909052925 | N/A | http://www.phenix-online.org/ |
| Software, algorithm | Chimera | https://doi.org/10.1002/jcc.20084 | N/A | https://www.cgl.ucsf.edu/chimera/ |
| Software, algorithm | PyMOL | Schrödinger | N/A | https://pymol.org/2/ |
| Software, algorithm | GraphPad Prism v7.0 | GraphPad Software | N/A | https://www.graphpad.com/ |
| Software, algorithm | FreeSASA | doi:10.12688/f1000research.7931.1 | N/A | http://freesasa.github.io/ |
| Software, algorithm | Gromacs 2018.5 | doi:10.1016/j.softx.2015.06.001 | N/A | https://manual.gromacs.org/2018.5/download.html |
| Software, algorithm | Protein Preparation Wizard | Schrödinger | N/A | https://www.schrodinger.com/products/protein-preparation-wizard |
| Software, algorithm | CHARMM-GUI Membrane Builder | https://doi.org/10.1002/jcc.23702 | N/A | https://charmm-gui.org/ |
| Software, algorithm | CHARMM36-CAMP | https://doi.org/10.1021/ct200328p | N/A | |
| Software, algorithm | LINCS algorithm | https://doi.org/10.1021/ct700200b | N/A | |
| Software, algorithm | Semi-isotropic Parrinello-Rahman barostat | https://doi.org/10.1016/0022-3093(93)90111-A | N/A | |

## Cell culture

*Spodoptera frugiperda* (*Sf*9) (Invitrogen) and High-Five insect cells (ThermoFisher Scientific) were cultured in ESF 921 serum-free medium (Expression Systems) at 27°C and 120 rpm.

## Constructs

The human GIPR DNA (Genewiz) with one mutation (T345F) was cloned into a modified pFastBac vector (Invitrogen). The native signal peptide was replaced by the hemagglutinin signal peptide (HA) to enhance receptor expression. A BRIL fusion protein was added at the N-terminal of the ECD with a TEV protease site and 2GSA linker between them. Forty-five amino acids (Q422-C466) were

truncated at the C terminus where LgBiT was added with a 15-amino acid (15AA) polypeptide linker in between, followed by a TEV protease cleavage site and an optimized maltose binding protein–maltose binding protein tag (OMBP-MBP). A dominant-negative bovine $G\alpha_s$ (DNG$\alpha_s$) (S54N, G226A, E268A, N271K, K274D, R280K, T284D, and I285T) construct was used to stabilize the complex (*Zhou et al., 2020*; *Liang et al., 2018b*). SmBiT34 (peptide 86, Promega) subunit was added to the C terminus of rat Gβ1 with a 15AA polypeptide linker between them. The modified rat Gβ1 and bovine Gγ2 were both cloned into a pFastBac vector.

### Protein expression

Baculoviruses containing the above complex construct were prepared by the Bac-to-Bac system (Invitrogen). GIPR and $G_s$ heterotrimer were co-expressed in High-Five cells. Briefly, insect cells were grown in ESF 921 culture medium (Expression Systems) to a density of $3.2 \times 10^6$ cells/mL, and then cells were infected with four kinds of viral preparations: BRIL-TEV-2GSA-GIPR(22-421)T345F-15AA-LgBiT-TEV-OMBP-MBP, $G\alpha_s$, Gβ1-peptide 86, and Gγ2 at a ratio of 1:3:3:3. After 48 hr incubation at 27°C, the cells were collected by centrifugation and stored at −80°C until use.

### Nb35 expression and purification

Nanobody-35 (Nb35) with a 6× his tag at the C terminus was expressed in the periplasm of *E. coli* BL21 (DE3) cells. Briefly, Nb35 target gene was transformed in the bacterium and amplified in TB culture medium with 100 μg/mL ampicillin, 2 mM $MgCl_2$, 0.1% (w/v) glucose at 37°C, 180 rpm. When OD600 reached 0.7–1.2, 1 mM IPTG was added to induce expression followed by overnight incubation at 28°C. The cell pellet was then collected at 3000 rpm under 4°C and stored at −80°C. Nb35 was purified as by size-exclusion chromatography using a HiLoad 16/600 Superdex 75 column (GE Healthcare) with running buffer containing 20 mM HEPES, 100 mM NaCl, pH 7.4. Fractions of Nb35 were concentrated to ~3 mg/mL and quickly frozen in the liquid nitrogen with 10% glycerol and stored in −80°C.

### Complex formation and purification

Cell pellets were lysed in a buffer consisting of 20 mM HEPES, 100 mM NaCl, pH 7.4, 10 mM $MgCl_2$, 1 mM $MnCl_2$, and 10% glycerol supplemented with protease inhibitor cocktail, EDTA-free (Traget-Mol). Subsequently, cell membranes were collected by ultracentrifugation at 4°C, 90,000 g for 35 min. The membranes were resuspended with a buffer containing 20 mM HEPES, 100 mM NaCl, pH 7.4, 10 mM $MgCl_2$, 1 mM $MnCl_2$, and 10% glycerol. The complex of GIPR-$G_s$ was assembled by adding 15 μM $GIP_{1-42}$ (GenScript), 100 μM TCEP, 25 mU/mL Apyrase (Sigma-Aldrich), 15 μg/mL Nb35, and 100 U salt active nuclease (Sigma-Aldrich) supplemented with protease inhibitor cocktail for 1.5 hr incubation at room temperature (RT). The preparation was then solubilized with 0.5% (w/v) lauryl maltose neopentylglycol (LMNG, Anatrace) and 0.1% (w/v) cholesterol hemisuccinate (CHS, Anatrace) with additional 1 μM $GIP_{1-42}$ for 3 hr at 4°C. The supernatant was isolated by centrifugation at 90,000 g for 35 min, and the solubilized complex was incubated with amylose resin (NEB) for 2.5 hr at 4°C. After batch binding, the resin was collected by centrifugation at 550 g and loaded onto a gravity flow column. The resin in column was firstly washed with five column volumes of buffer containing 20 mM HEPES, pH 7.4, 100 mM NaCl, 10% (v/v) glycerol, 5 mM $MgCl_2$, 1 mM $MnCl_2$, 25 μM TCEP, 3 μM $GIP_{1-42}$, 0.1% (w/v) LMNG, and 0.02% (w/v) CHS. Subsequently, the resin was washed with 25 column volumes of buffer containing 20 mM HEPES, pH 7.4, 100 mM NaCl, 10% (v/v) glycerol, 5 mM $MgCl_2$, 1 mM $MnCl_2$, 25 μM TCEP, 3 μM $GIP_{1-42}$, 0.03% (w/v) LMNG, 0.01% (w/v) glyco-diosgenin (GDN, Anatrace), and 0.008% (w/v) CHS. The protein was then incubated with a buffer containing 20 mM HEPES, pH 7.4, 100 mM NaCl, 10% (v/v) glycerol, 5 mM $MgCl_2$, 1 mM $MnCl_2$, 25 μM TCEP, 50 μM $GIP_{1-42}$, 10 μg/mL Nb35, 0.03% (w/v) LMNG, 0.01% (w/v) glyco-diosgenin, 0.008% (w/v) CHS, and 30 μg/mL His-tagged TEV protease on the column overnight at 4°C. The flow through was collected and concentrated to 500 μL using a 100 kDa filter (Merck Millipore). Size-exclusion chromatography was performed by loading the protein onto Superose 6 Increase 10/300 GL (GE Healthcare) column with running buffer containing 20 mM HEPES, pH 7.4, 100 mM NaCl, 10 mM $MgCl_2$, 100 μM TCEP, 5 μM $GIP_{1-42}$, 0.00075% (w/v) LMNG, 0.00025% (w/v) glyco-diosgenin, 0.0002% (w/v) CHS, and 0.00025% digitonin (Anatrace). Monomeric GIPR-$G_s$ complexes were collected and concentrated for cryo-EM analysis.

## Data acquisition and image processing

The purified GIP$_{1-42}$–GIPR–G$_s$–Nb35 complex at a concentration of 6–7 mg/mL was mixed with 100 µM GIP$_{1-42}$ at 4°C and applied to glow-discharged holey carbon grids (Quantifoil R1.2/1.3, Au 300 mesh) that were subsequently vitrified by plunging into liquid ethane using a Vitrobot Mark IV (ThermoFisher Scientific). A Titan Krios equipped with a Gatan K3 Summit direct electron detector was used to acquire Cryo-EM images. The microscope was operated at 300 kV accelerating voltage, at a nominal magnification of 46,685× in counting mode, corresponding to a pixel size of 1.071 Å. Totally, 8023 movies were obtained with a defocus range of −1.2 to −2.2 µm. An accumulated dose of 80 electrons per Å$^2$ was fractionated into a movie stack of 36 frames.

Dose-fractionated image stacks were subjected to beam-induced motion correction using MotionCor2.1. A sum of all frames, filtered according to the exposure dose, in each image stack was used for further processing. Contrast transfer function parameters for each micrograph were determined by Gctf v1.06. Particle selection, 2D and 3D classifications were performed on a binned dataset with a pixel size of 2.142 Å using RELION-3.0-beta2. Auto-picking yielded 4,895,399 particle projections that were subjected to reference-free 2D classification to discard false-positive particles or particles categorized in poorly defined classes, producing 2,754,623 particle projections for further processing. This subset of particle projections was subjected to a round of maximum-likelihood-based three dimensional classifications with a pixel size of 2.142 Å, resulting in one well-defined subset with 1,395,031 projections. Further 3D classifications with mask on the receptor produced one good subset accounting for 565,239 particles, which were subjected to another round of 3D classifications with mask on the ECD. A selected subset containing 295,021 projections was then subjected to 3D refinement and Bayesian polishing with a pixel size of 1.071 Å. After the last round of refinement, the final map has an indicated global resolution of 2.94 Å at a Fourier shell correlation (FSC) of 0.143. Local resolution was determined using the Bsoft package with half maps as input maps.

## Model building and refinement

The cryo-EM structure of GCGR–G$_s$–Nb35 complex (PDB code 6WPW) (*Qiao et al., 2020*) and the crystal structure of GIPR ECD (PDB code 2QKH) (*Parthier et al., 2007*) were used as the start for model building and refinement against the EM map. The model was docked into the EM density map using Chimera (*Pettersen et al., 2004*), followed by iterative manual adjustment and rebuilding in COOT (*Emsley and Cowtan, 2004*). Real space refinement was performed using Phenix (*Adams et al., 2010*). The model statistics were validated using MolProbity (*Chen et al., 2010*). Structural figures were prepared in Chimera and PyMOL (https://pymol.org/2/). The final refinement statistics are provided in *Table 1*.

## cAMP accumulation assay

GIP$_{1-42}$-stimulated cAMP accumulation was measured by a LANCE Ultra cAMP kit (PerkinElmer). Briefly, HEK 293T cells were cultured in DMEM (Gibco) supplemented with 10% (v/v) fetal bovine serum (FBS, Gibco) and 1% (v/v) sodium pyruvate (Gibco) at 37°C, 5% CO$_2$. Cells were seeded onto six-well cell culture plates and transiently transfected with different GIPR constructs using Lipofectamine 2000 transfection reagent (Invitrogen). All the mutant constructs were modified by single-point mutation in the setting of the WT construct (HA-Flag-3GSA-GIPR(22-466)). After 24 hr culture, the transfected cells were seeded onto 384-well microtiter plates at a density of 3000 cells per well in HBSS supplemented with 5 mM HEPES, 0.1% (w/v) bovine serum albumin (BSA), and 0.5 mM 3-isobutyl-1- methylxanthine. The cells were stimulated with different concentrations of GIP$_{1-42}$ for 40 min at RT. Eu and Ulight were then diluted by cAMP detection buffer and added to the plates separately to terminate the reaction. Plates were incubated at RT for 40 min and the fluorescence intensity measured at 620 nm and 650 nm by an EnVision multilabel plate reader (PerkinElmer).

## Whole-cell binding assay

CHO-K1 cells were cultured in F12 medium with 10% FBS and seeded at a density of 30,000 cells/well in Isoplate-96 plates (PerkinElmer). The WT (HA-Flag-3GSA-GIPR(22-466)) or mutant GIPR were transiently transfected using Lipofectamine 2000 transfection reagent. The mutant construct was modified by single-point mutation in the setting of the WT construct. Twenty-four hours after transfection, cells were washed twice, and incubated with blocking buffer (F12 supplemented with 33

mM HEPES and 0.1% BSA, pH 7.4) for 2 hr at 37℃. For homogeneous binding, cells were incubated in binding buffer with a constant concentration of $^{125}$I-GIP (40 pM, PerkinElmer) and increasing concentrations of unlabeled GIP$_{1-42}$ (3.57 pM–1 µM) at RT for 3 hr. Following incubation, cells were washed three times with ice-cold PBS and lysed by addition of 50 µL lysis buffer (PBS supplemented with 20 mM Tris–HCl, 1% Triton X-100, pH 7.4). Fifty microliters of scintillation cocktail (OptiPhase SuperMix, PerkinElmer) was added, and the plates were subsequently counted for radioactivity (counts per minute, CPM) in a scintillation counter (MicroBeta2 Plate Counter, PerkinElmer).

### β-Arrestin2 recruitment

HEK 293T cells (3 × 10$^6$ cells/10 cm plate) were grown for 24 hr before transfection with 10.6 µg plasmid containing GIPR tagged with Rluc8 and β-arrestin with a Venus-tag in the N terminus at a ratio of 1:9. Transiently transfected cells were then seeded onto poly-D-lysine coated 96-well culture plates (50,000 cells/well) in DMEM with 10% FBS. Cells were grown overnight before incubation in assay buffer (HBSS supplemented with 10 mM HEPES and 0.1% BSA, pH 7.4) for 30 min at 37℃. Coelentrazine-h (Yeasen Biotech) was added to a final concentration of 5 µM for 5 min before bioluminescence resonance energy transfer (BRET) readings were made using an EnVision plate reader (PerkinElmer). BRET baseline measurements were collected for 10 cycles prior to ligand addition. Following peptide addition, BRET was measured for 50 cycles. The BRET signal (ratio of 535 nm over 470 nm emission) was corrected to the baseline and then vehicle-treated condition to determine ligand-induced changes in BRET response. Concentration–response values were obtained from the area-under-the-curve (AUC) of the responses elicited by GIP$_{1-42}$.

### Receptor surface expression

Cell surface expression was determined by flow cytometry to the N-terminal Flag tag on the WT GIPR (HA-Flag-3GSA-GIPR(22-466)) and its mutants transiently expressed in HEK 293T cells. All the mutant constructs were modified by single-point mutation in the setting of the WT construct. Briefly, approximately 2 × 10$^5$ cells were blocked with PBS containing 5% BSA (w/v) at RT for 15 min and then incubated with 1:300 anti-Flag primary antibody (diluted with PBS containing 5% BSA, Sigma-Aldrich) at RT for 1 hr. The cells were then washed three times with PBS containing 1% BSA (w/v) followed by 1 hr incubation with 1:1000 anti-mouse Alexa Fluor 488 conjugated secondary antibody (diluted with PBS containing 5% BSA, Invitrogen) at RT in the dark. After washing three times, cells were re-suspended in 200 µL PBS containing 1% BSA for detection by NovoCyte (Agilent) utilizing laser excitation and emission wavelengths of 488 nm and 530 nm, respectively. For each sample, 20,000 cellular events were collected, and the total fluorescence intensity of positive expression cell population was calculated. Data were normalized to the WT receptor.

### Molecular dynamics simulations

Molecular dynamic simulations were performed by Gromacs 2018.5. The peptide–GIPR complexes were built based on the cryo-EM GIP–GIPR–G$_s$ complex and prepared by the Protein Preparation Wizard (Schrodinger 2017–4) with the G protein and Nb35 nanobody removed. The receptor chain termini were capped with acetyl and methylamide, and the titratable residues were left in their dominant state at pH 7.0. The complexes were embedded in a bilayer composed of 200 POPC lipids and solvated with 0.15 M NaCl in explicitly TIP3P waters using CHARMM-GUI Membrane Builder (*Wu et al., 2014*). The CHARMM36-CAMP force filed (*Guvench et al., 2011*) was adopted for protein, peptides, lipids, and salt ions. The Particle Mesh Ewald (PME) method was used to treat all electrostatic interactions beyond a cut-off of 10 Å, and the bonds involving hydrogen atoms were constrained using LINCS algorithm (*Hess, 2008*). The complex system was firstly relaxed using the steepest descent energy minimization, followed by slow heating of the system to 310 K with restraints. The restraints were reduced gradually over 50 ns. Finally, restrain-free production run was carried out for each simulation, with a time step of 2 fs in the NPT ensemble at 310 K and 1 bar using the Nose–Hoover thermostat and the semi-isotropic Parrinello–Rahman barostat (*Aoki and Yonezawa, 1992*), respectively. The buried interface areas were calculated with FreeSASA (*Mitternacht, 2016*) using the Sharke–Rupley algorithm with a probe radius of 1.2 Å. The last 700 ns trajectory of each simulation was used to root mean square fluctuation (RMSF) calculation.

## Statistical analysis

All functional data were presented as means ± standard error of the mean (S.E.M.). Statistical analysis was performed using GraphPad Prism 7 (GraphPad Software). Concentration–response curves were evaluated with a three-parameter logistic equation. The significance was determined with either two-tailed Student's $t$-test or one-way ANOVA. Significant difference is accepted at $p < 0.001$.

## Acknowledgements

We thank Elita Yuliantie, Wen Sun, Zhaotong Cong, Fulai Zhou, Yuqi Ping, X Edward Zhou, Karsten Melcher, Jinhuan Chen, and Xijiang Pan for technical advice. The cryo-EM data were collected at Cryo-Electron Microscopy Research Center, Shanghai Institute of Materia Medica. This work was partially supported by National Natural Science Foundation of China 81872915 (M-WW), 32071203 (LHZ), 81773792 (DHY), 81973373 (DHY), and 21704064 (QTZ); National Science and Technology Major Project of China – Key New Drug Creation and Manufacturing Program 2018ZX09735–001 (M-WW) and 2018ZX09711002–002–005 (DHY); National Key Basic Research Program of China 2018YFA0507000 (M-WW); Ministry of Science and Technology of China 2018YFA0507002 (HEX); Shanghai Municipal Science and Technology Major Project 2019SHZDZX02 (HEX); Strategic Priority Research Program of Chinese Academy of Sciences XDB37030103 (HEX); Shanghai Municipality Science and Technology Development Fund 18430711500 (M-WW) and 18ZR1447800 (LHZ); Novo Nordisk-CAS Research Fund grant NNCAS-2017–1-CC (DHY); The Young Innovator Association of CAS 2018325 (LHZ); and SA-SIBS Scholarship Program (LHZ and DHY).

## Additional information

### Funding

| Funder | Grant reference number | Author |
| --- | --- | --- |
| National Natural Science Foundation of China | 81872915 | Ming-Wei Wang |
| National Natural Science Foundation of China | 32071203 | Lihua Zhao |
| National Natural Science Foundation of China | 81773792 | Dehua Yang |
| National Natural Science Foundation of China | 81973373 | Dehua Yang |
| National Natural Science Foundation of China | 21704064 | Qingtong Zhou |
| National Science and Technology Major Project of China | 2018ZX09735-001 | Ming-Wei Wang |
| National Science and Technology Major Project of China | 2018ZX09711002-002-005 | Dehua Yang |
| National Key Basic Research Program of China | 2018YFA0507000 | Ming-Wei Wang |
| Ministry of Science and Technology of China | 2018YFA0507002 | H Eric Xu |
| Shanghai Municipal Science and Technology Commission | 2019SHZDZX02 | H Eric Xu |
| Chinese Academy of Sciences | XDB37030103 | H Eric Xu |
| Shanghai Science and Technology Development Fund | 18430711500 | Ming-Wei Wang |
| Novo Nordisk | NNCAS-2017-1-CC | Dehua Yang |
| Shanghai Science and Technology Development Fund | 18ZR1447800 | Lihua Zhao |
| Youth Innovation Promotion | 2018325 | Lihua Zhao |

Association of the Chinese
Academy of Sciences

Shanghai Institutes for Biologi-
cal Sciences, Chinese Academy
of Sciences                                         Dehua Yang
                                                    Lihua Zhao

The funders had no role in study design, data collection and interpretation, or the
decision to submit the work for publication.

## Author contributions
Fenghui Zhao, Data curation, Software, Formal analysis, Validation, Visualization, Methodology, Writ-
ing - original draft; Chao Zhang, Yan Chen, Data curation, Formal analysis, Validation, Methodology;
Qingtong Zhou, Xinyu Zou, Software, Formal analysis, Visualization, Methodology, Writing - original
draft; Kaini Hang, Data curation, Formal analysis, Methodology; Fan Wu, Wanchao Yin, Dan-Dan
Shen, Yan Zhang, Methodology; Qidi Rao, Data curation; Antao Dai, Data curation, Validation, Meth-
odology; Tian Xia, Software; Raymond C Stevens, Conceptualization; H Eric Xu, Conceptualization,
Resources, Supervision, Funding acquisition, Investigation, Project administration, Writing - review
and editing; Dehua Yang, Resources, Data curation, Formal analysis, Supervision, Funding acquisi-
tion, Validation, Investigation, Methodology, Writing - original draft, Project administration, Writing -
review and editing; Lihua Zhao, Resources, Data curation, Formal analysis, Supervision, Funding
acquisition, Investigation, Methodology, Writing - original draft, Writing - review and editing; Ming-
Wei Wang, Conceptualization, Resources, Formal analysis, Supervision, Funding acquisition, Investi-
gation, Methodology, Writing - original draft, Project administration, Writing - review and editing

## Author ORCIDs
Qingtong Zhou [iD] https://orcid.org/0000-0001-8124-3079
Dehua Yang [iD] https://orcid.org/0000-0003-3028-3243
Ming-Wei Wang [iD] https://orcid.org/0000-0001-6550-9017

## Decision letter and Author response
Decision letter https://doi.org/10.7554/eLife.68719.sa1
Author response https://doi.org/10.7554/eLife.68719.sa2

# Additional files

## Supplementary files
• Source data 1. All the source data files.

• Transparent reporting form

## Data availability
Atomic coordinates of the GIP-GIPR-$G_s$ complex have been deposited in the Protein Data Bank
under accession code 7DTY and Electron Microscopy Data Bank (EMDB) accession code EMD-
30860. All data generated or analysed during this study are included in the manuscript and support-
ing files. Source data files have been provided for Figure 2, Figure 1—figure supplement 1 and Fig-
ure 4—figure supplement 4.

The following dataset was generated:

| Author(s) | Year | Dataset title | Dataset URL | Database and Identifier |
|---|---|---|---|---|
| Zhao F, Zhang C, Zhou Q, Hang K, Zou X, Chen Y, Wu F, Rao Q, Dai A, Yin W, Shen DD, Zhang Y, Xia T, Stevens RC, Xu HE, Yang D, Zhao L, Wang MW | 2021 | Structural basis of ligand selectivity conferred by the human glucose-dependent insulinotropic polypeptide receptor | https://www.ebi.ac.uk/pdbe/entry/emdb/EMD-30860 | Electron Microscopy Data Bank, EMD-30 860 |

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
