## [Decision Letter]

**Acceptance summary:**

The paper describes the cryo-EM structure of the class B receptor glucose-dependent insulinotropic polypeptide (GIP)-receptor in complex with the Gs heterotrimer and the 42 amino acid hormone GIP, which is an incretin and stimulates insulin secretion in a glucose-dependent manner. The complex structure provides important insights into ligand recognition, receptor activation and transducer coupling. In particular, the interaction of GIP with GIPR reveals previously unresolved details of how the hormone is recognized by the receptor, and the structural differences compared to the previously obtained structure of the incretin glucagon-like peptide 1-Gs complex.

**Decision letter after peer review:**

Thank you for submitting your article "Structural insights into hormone recognition by the human glucose-dependent insulinotropic polypeptide receptor" for consideration by *eLife*. Your article has been reviewed by 2 peer reviewers, and the evaluation has been overseen by a Reviewing Editor and Volker Dötsch as the Senior Editor. The following individual involved in review of your submission has agreed to reveal their identity: Arun Shukla (Reviewer #2).

Essential revisions:

1) As the authors have pointed out that map density for some of the ECD was poor. Currently the structure for the ECD has been modelled from previously determined structures of closely related GLP/GCGR. However, after inspection in Coot these models actually do not fit part of the ECD density that is present, indicating some differences here. Furthermore, detailed interactions are discussed in the paper between the ECD and GIP peptide that cannot be validated by the present maps as the location of some the side-chains are ambiguous. It is essential that the authors either I) improve the current cryo EM maps and/or re-model the structure to more faithfully fit the map density. In this regard, some further experimental and/or computational support might be needed if it is not possible to improve the map quality. The text should be appropriately adjusted.

2) It would be helpful to clarify how significant is the ambiguity in the C-terminal end of the modelled GIP peptide? This raises an important point since only 30 out of the 42 GIP residues could be modelled. Is 30aa of the GIP peptide enough for full activation? The GLP-1 is only 30 or 31 amino acids long. Is it known why the GIP peptide is longer as is the length of this peptide important for specificity, if so, then the poorly modelled region at the end of the peptide might be significant and should be clarified.

---

## [Author Response]

Essential revisions:1) As the authors have pointed out that map density for some of the ECD was poor. Currently the structure for the ECD has been modelled from previously determined structures of closely related GLP/GCGR. However, after inspection in Coot these models actually do not fit part of the ECD density that is present, indicating some7 differences here. Furthermore, detailed interactions are discussed in the paper between the ECD and GIP peptide that cannot be validated by the present maps as the location of some the side-chains are ambiguous. It is essential that the authors either I) improve the current cryo EM maps and/or re-model the structure to more faithfully fit the map density. In this regard, some further experimental and/or computational support might be needed if it is not possible to improve the map quality. The text should be appropriately adjusted.

We appreciate the reviewers’ insightful comments. Compared to the core structure, the density of ECD is relatively poor owning to its intrinsic flexibility, which is a general feature in most reported class B1 GPCR–G_s_ complex structures. Based on an X-ray crystal structure of GIPR ECD (PDB code: 2QKH), the initial ECD model was built and then docked into the cryo-EM density with necessary modifications, where the ECD model may not be entirely fitted to the density in several regions.

To this point, we have gone through the model and adjusted several regions of the ECD to better fit the map density. The loop residues 104 through 111 in the ECD are omitted due to poor densities. We further refined the model with the rosetta refinement program (phenix.rosetta_refine) to improve the protein geometry and rotamer assignments, especially for those residues in the ECD region. These efforts have significantly improved the quality of our ECD model.

The descriptions of interactions between the ECD and GIP have thus been revised accordingly reflecting our mutagenesis data (Figure 2D) showing mutations in the intact or truncated ECDs greatly reduced the potency of GIP. “The C-terminal half of GIP was clasped by the GIPR ECD, closely resembling the crystal structure of GIP–GIPR ECD (PDB code: 2QKH) (*21*). Consistent with the interaction patterns observed in other class B1 GPCRs (14-17), Specifically, three adjacent amino acids (H18^P^, Q19^P^ and Q20^P^) form multiple hydrogen bonds with the side-chains of Q30 and Y36 at the N-terminal α-helix of ECD, as well as N120 and E122 at the hinge region between ECD and TM1. The hydrophobic residues of GIP (F22^P^, V23^P^, L26^P^ and L27^P^) in the C-terminal half of GIP occupy a complementary binding groove of the GIPR ECD, consisting of a series of hydrophobic residues (L35, Y36, W39, M67, Y68, Y87, L88, P89 and W90). Alanine substitutions in W39, D66 and Y68 significantly reduced the potency of GIP (Figure 2D). Besides, several polar contacts including H18^P^-Y36 and Q20^P^-N124 were observed.”

2) It would be helpful to clarify how significant is the ambiguity in the C-terminal end of the modelled GIP peptide? This raises an important point since only 30 out of the 42 GIP residues could be modelled. Is 30aa of the GIP peptide enough for full activation? The GLP-1 is only 30 or 31 amino acids long. Is it known why the GIP peptide is longer as is the length of this peptide important for specificity, if so, then the poorly modelled region at the end of the peptide might be significant and should be clarified.

We thank the reviewers for the comments. GIP is initially produced as a 153-amino acid pro-hormone that is processed to become GIP(1-42) and GIP(1-30)NH_2_, while the former is the main active form of GIP, and the latter is also active in the circulation. According to NMR data and the crystal structure of GIPR ECD (PDB code: 2QKH), the last 12 residues of GIP(1–42) are disordered and neither bind to the ECD nor to the TMD, indicating that these residues are not of important for ligand recognition and receptor activation (PMID: 31809770). Consistently, GIP(1–42) and GIP(1–30)NH_2_ bind with almost identical affinity (Author response image 1) to the receptor thereby confirming the redundancy of residues Gly31 –Gln42 (PMID: 31809770, 20185691, 26572091 and 32360365, Author response image 2).

**Author response image 1. sa2fig1:** cAMP signaling profiles of GIP(1–42) and GIP(1–30)NH_2_. (PMID: 31809770).

**Author response image 2. sa2fig2:** Pharmacological profiles of the naturally occurring GIPR agonists, GIP(1–42) and GIP(1–30)NH_2_. (A) Competition of ^125^I-GIP(1-42) binding to GIPR by GIP analogues. Binding affinity data are expressed as a percentage of measured bound *vs.* bound in the absence of peptide, each corrected for non-specific binding (measured in the presence of 1 μM of GIP(1-42) or GLP-1(7-36)NH_2_). (B) Gα_s_ activation in GIPR expressing cells by GIP analogues. Gα_s_ activation assay was performed in HEK293A cells transiently transfected with either GIPR or GLP-1R. (C) cAMP accumulation at GIPR by GIP analogues. The assay was performed in HEK293 cells stably expressing GIPR or GLP-1R. Measurement was converted to absolute cAMP levels using a standard curve and then normalized to the maximal response of the cognate ligands for each receptor. (D) β-arrestin2 (β-arr2) recruitment to GIPR by GIP analogues. β-arrestin recruitment assay was performed in HEK293T cells transiently transfected with either Rluc8 tagged GIPR and Venus tagged β-arrestin 2. Data of at least four independent experiments are fitted to nonlinear regression three-parameter logistic curves. All values are means ± S.E.M. (PMID: 32360365).